# Basin-scale sources and pathways of microplastic that ends up in the Galápagos Archipelago

Erik van Sebille[1], Philippe Delandmeter[1], John Schofield[2], Britta Denise Hardesty[3], Jen Jones[4,5], and Andy Donnelly[4]

[1]Institute for Marine and Atmospheric research, Utrecht University, Utrecht, Netherlands
[2]Department of Archeology, University of York, York, United Kingdom
[3]Commonwealth Scientific and Research Organization, Oceans and Atmosphere, Hobart, TAS, Australia
[4]Galapagos Conservation Trust, London, United Kingdom
[5]University of Exeter, Exeter, United Kingdom

**Correspondence:** Erik van Sebille (E.vanSebille@uu.nl)

**Abstract.** The Galápagos Archipelago and Marine Reserve lies 1,000km off the coast of Ecuador and is among the world's most iconic wildlife refuges. However, plastic litter is now found even in this remote island archipelago. Prior to this study, the sources of this plastic litter on Galápagos coastlines were unidentified. Local sources are widely expected to be small, given the limited population and environmentally-conscious tourism industry. Here, we show that remote sources of plastic pollution are also fairly localized and limited to nearby fishing regions and South and Central American coastlines, in particular Northern Peru and Southern Ecuador. Using virtual floating plastic particles transported in high-resolution ocean surface currents, we analysed the plastic origin and fate using pathways and connectivity between the Galápagos region and the coastlines and known fisheries locations around the East Pacific Ocean. We also analysed how incorporation of wave-driven currents (Stokes drift) affects these pathways and connectivity. We found that only virtual particles that enter the ocean from Peru, Ecuador and (when waves are not taken into account) Colombia can reach the Galápagos. It takes these particles a few months to travel from their coastal sources on the American continent to the Galápagos region. The connectivity does not seem to vary substantially between El Niño and La Niña years. Identifying these sources and the timing and patterns of the transport can be useful for identifying integrated management opportunities to reduce plastic pollution from reaching the Galápagos Archipelago.

## 1 Introduction

Marine plastic litter has in only a few decades become ubiquitous in our oceans (e.g. Law, 2017). Plastic is now found in even the most remote locations, including the deep seafloor (Woodall et al., 2014), uninhabited islands (Lavers and Bond, 2017), in the Arctic (Cózar et al., 2017) and in the waters around and coastlines of Antarctica (Waller et al., 2017). Yet, there are significant spatial differences in the concentration of plastic. On the surface of the ocean, for example, the estimated

concentration of small floating plastic is 10 million times higher in the subtropical accumulation regions than in the Southern Ocean (van Sebille et al., 2015). Because of deep upwelling of water in the Southern Ocean and Ekman drift towards the subtropical gyres (Rintoul and Naveira Garabato, 2013), there is a net transport of floating plastic away from the region (Onink et al., 2019). The same is true for regions on the Equator, such as the Galápagos Archipelago, where upwelling and surface
divergence mean that the surface flow is predominantly directed away from the Equator (Law et al., 2014).

The Galápagos Archipelago and Marine Reserve are among the world's most valued and most iconic ecosystems. Its special qualities were first noticed when Charles Darwin visited the archipelago in 1835. They were later recognised in the islands being granted the first UNESCO World Heritage status for natural value in 1978, with the marine reserve following the archipelago itself onto the UNESCO World Heritage List two decades later. However, even this remote archipelago is not as pristine as one
would hope (Mestanza et al., 2019). So, despite the archipelago being in a region of ocean surface divergence (Fiedler et al., 1991) with relatively low expected plastic concentrations, the blight of plastic pollution has now also arrived in Galápagos. There, it has unquantified but likely significant impacts on the unique ecosystem as well as on the sustainability of the tourism industry which supports the economy of the Galápagos locally, and Ecuador more broadly.

Management and mitigation of the plastic problem in the Galápagos Archipelago requires understanding the scale and
sources of the pollution. While some of the plastic found on coastlines and in the marine reserve may originate from the islands themselves, including tourism, there is a widespread view, based on information from coastal clean up efforts (Galápagos National Park, unpublished data), that much of the plastic found in the Galápagos comes from mainland America, from continental Asia, and from fisheries in the Pacific Ocean.

Here, we investigated the pathways of floating microplastic between the Galápagos Islands and coastlines and known fish-
eries locations around the Pacific. There is some observational data on pathways into the Galápagos region, from satellite-tracked surface drifters in the real ocean. However, of the more than 30,000 drifters in the Global Drifter Program (GDP) (Elipot et al., 2016), only 40 crossed the Galápagos Archipelago region, defined as between [91.8°W - 89°W, 1.4°S - 0.7°N] (Figure 1). Most of these 40 drifters were released relatively close to the Galápagos, in the Eastern Tropical Pacific Ocean (Figure 1 upper panel). After leaving the Galápagos region, many of the drifters crossed the entire Pacific Ocean. Very clear
here is the divergent flow at the Equator, where the drifters move poleward on both Hemispheres (Figure 1, lower panel).

To augment the GDP drifter observations, we employ state-of-the-art numerical models. We used a combination of the fine-resolution NEMO global hydrodynamic model for ocean surface currents (Madec, 2008), the WaveWatch III model for waves (Tolman, 2009), and the Parcels v2.0 Lagrangian particle tracking toolbox (Lange and van Sebille, 2017; Delandmeter and van Sebille, 2019). We compared these with the trajectories of floating drifters in the real ocean.
There is still a debate in the physical oceanography community to what extent wave-induced currents – so-called Stokes drift (Stokes, 1847) – has an impact on the transport of plastic (Lebreton et al., 2018; Onink et al., 2019). Therefore, we analyzed the particle pathways both with and without this effect of waves. Stokes drift is the net drift velocity in the direction of wave propagation experienced by a particle floating at the free surface of a water wave (see van den Bremer and Breivik, 2018, for a recent review). Its magnitude is generally much smaller than that of the surface currents (e.g. Figure 1 of Onink et al., 2019),

but because Stokes drift has large spatial coherence patterns its long-term effect on particle transport can be significant (Fraser et al., 2018).

Finally, we also describe how the modelling performed here can work alongside other methodologies, to demonstrate the benefits of multidisciplinary approaches to helping resolve the problem of marine plastic pollution.

## 2 Methods

We performed six experiments in three scenarios: one scenario where we tracked the origin of particles, by computing particles that end up near the Galápagos in backward time; one scenario where we tracked the fate of particles that were released from the west coast of the Americas in forward time; and one scenario where we tracked the fate of particles that were released at know fishing locations in forward time. In all three scenarios, we simulated the transports by ocean surface currents only and by the combination of surface currents and waves. As the NEMO model data is available on 8km resolution, we focused only on the basin-scale transports, and leave transports within and between the different islands of the Galápagos Archipelago for future work.

We used the two-dimensional surface flow fields from the NEMO hydrodynamic model, simulation ORCA0083-N006, which has a global coverage at 1/12° resolution (nominally 8km around the Equator) (Madec, 2008). The NEMO data is available from January 2000 to December 2010, on 5-day temporal resolution. As Qin et al. (2014) showed that time-averaging errors are small for temporal resolutions shorter than 9 days in a 1/10° spatial resolution, this 5-day temporal resolution is sufficient.

For the Stokes currents, we used the WaveWatch III data based on CFSR winds (Tolman, 2009), which has a global coverage at 1/2° resolution (nominally 55km around the Equator). The WaveWatch III data is also available from January 2000 to December 2010, on 3-hour temporal resolution.

We advected Lagrangian particles using the Parcels v2.0 toolbox (Lange and van Sebille, 2017; Delandmeter and van Sebille, 2019) in either only the NEMO surface flow fields (hereafter referred to as the 'currents' simulations), or the combined NEMO surface flow and WaveWatchIII Stokes drift fields (hereafter referred to as the 'currents+waves' simulations). Parcels v2.0 has inbuilt support for advection of particles on multiple different Fields using SummedField objects, so that the velocities at each location are interpolated and then summed at each RK4 substep (see also Delandmeter and van Sebille, 2019), and the currents+wave simulations were performed using that feature. The particles represented microplastic that are sufficiently buoyant to not mix too deep in the mixed layer (Onink et al., 2019). We used a Runge-Kutta 4 integration scheme with a time-step of one hour. We stored the location of each particle on a daily (24 hours) resolution. All scripts that were used to run the simulations are available at https://github.com/OceanParcels/GalapagosBasinPlastic.

On each set of fields, we performed three different simulations based on three scenarios. In the 'Origin from Galápagos' scenario, we released 154 particles every 10 days in a box between [91.8°W - 89°W, 1.4°S - 0.7°N] (the red box in Figure 2), on a 0.2°x0.2° grid, for a total of 61,908 particles. We integrated these particles back in time for a maximum length of 10

years, or until the first day available in the NEMO dataset. Redoing all the analyses below with only half of the particles does not affect the results and conclusions, giving us confidence that we released sufficient particles.

In the 'Fate from the South American coastline' scenario, we released one particle each 0.5° between 38°S and 31°N every 5 days, for a total of 120,450 particles. Again, using only half of the particles in our analysis did not change the results and conclusions drawn below. For each latitude, we picked the easternmost longitude that is still in the Pacific Ocean, so that the release points traced the coastline of the Americas. We then integrated our particles forward in time for a maximum of 5 years, or until the last day available in the NEMO dataset. We identified those particles that crossed the box at [91.8°W - 89°W, 1.4°S - 0.7°N], the same box as the release for the 'Origin from Galápagos' simulation, and defined these to be passing through the Galápagos Archipelago region.

In the 'Fate from regional fisheries' scenario, we released particles according to the distribution of total fishing effort, as mapped by Global Fishing Watch (Kroodsma et al., 2018), in a region around the Galápagos (Figure 2). We selected only these locations where there was at least 24 hours of fishing activity between 1 January 2012 and 31 December 2016. As these dates did not overlap with the available NEMO surface flow data from 2000 to 2010, we repeatedly released one particle each month - weighted to the number of fishing hours - at each of the 3,885 locations in Figure 2, for a total of 520,590 particles. We then integrated these particles forward in time for a maximum of 5 years, or until the last day available in the NEMO dataset. We used the same definition of passing through the Galápagos Archipelago region as in the 'Fate from the South American coastline' simulations above.

## 3   Results

In the 'Origin from Galápagos' scenario, most particle trajectories were confined to the Eastern Tropical Pacific Ocean, the South American coastline, and the Antarctic Circumpolar Current (Figure 3). In the currents+waves run, some particles even arrived in the Galápagos region that originated from the Indian Ocean (Maes et al., 2018; van der Mheen et al., 2019). However, none of the almost 65,000 particles came from the North or South Pacific accumulation zones (Kubota, 1994; Martinez et al., 2009; Eriksen et al., 2013; van Sebille et al., 2015) or from even close to mainland Asia. While some particles in the currents-only simulation originated from the very southern part of California, most particles originated from much farther south. Interestingly, the inclusion of Stokes drift meant that particles were much more dispersed through the Southern Ocean, in agreement with recent simulations of Kelp in that region (Fraser et al., 2018).

In the 'Fate from the South American coastline' scenario, most particles released from the American coastline ended up in either the North Pacific or South Pacific accumulation zones within the five years that they were advected for (Figure 4). Some particles even ended up in the Indian Ocean, having passed through the Indonesian Throughflow (e.g. van Sebille et al., 2014). There was a local minimum in the density of particle trajectories on the Equator, especially west of the Galápagos, which agrees with the GDP drifters (lower panel of Figure 1). Compared to the currents-only simulation, the convergence zones were more spread-out and reached farther westward in the currents+waves simulation. The accumulation zones were also smaller and had lower maxima in the currents+waves simulation, partly because the waves constantly push particles eastward onto the

shore, so that they had less chance of reaching the open ocean. Indeed, the narrow strip of very high concentrations seen along the South America coastline in the lower panel of Figure 4 confirms that one effect of the eastward Stokes drift induced by the waves was to contain the particles close to their release locations.

The fraction of particles that reached the Galápagos region, starting from the western American coast, is shown in Figure 5.
Only very few of the particles released south of 16°S or north of 3°N reached the Galápagos, and even for the regions between 16°S and 3°N the fraction of particles arriving in the Galápagos region is never higher than 25%. There was a clear difference between the two flow simulations: in the currents+waves simulation (blue line in Fig 5) the particles that reached the Galápagos came almost exclusively from Peru, while in the currents-only simulation there was also a significant fraction of virtual particles from Ecuador, Colombia, Costa Rica and even farther north.

In both 'Fate from the South American coastline' simulations, less than 1% of the particles from the Chilean coast arrived in the Galápagos region, even though in the 'Origin from Galápagos' scenario there was a clear pathway along the Chilean coast. This apparent inconsistency between the two scenarios is due to the fact that the interpretation of the origin and fate simulations is very different. Most of the particles that enter the ocean from the American coastline do not come close to the Galápagos region. However, in the 'Origin from Galápagos' simulation we tracked only those that do; so by construction they
all end there. This shows that forward and backward simulations can yield complementary information, even if the simulation of individual particles first forward in time and then backward in time returns them to their original position when the time-step goes to zero (e.g. Qin et al., 2014; van Sebille et al., 2018).

The travel time from the west coast of Americas to the Galápagos was typically a few months (Figure 6). In the currents+waves simulation, almost all particles that reached the Galápagos did so within 3 months (100 days; blue bars in Fig-
ure 6). In the currents-only simulation, there was a much longer tail, reaching travel times up to 5 years (yellow bars). Note however, that none of the simulations here take sinking of particles into account, which can be expected to be more likely for longer times at sea (Kooi et al., 2017; Koelmans et al., 2017). Furthermore, longer residence times in the ocean will also likely lead to more fragmentation, but this is also not taken into account because the time scales involved are very poorly constrained from observations (Cózar et al., 2014).

An analysis of the particles reaching the Galápagos from mainland America for each year showed that there was little impact of El Niños and La Niñas on the transport of particles from the American coastline to the Galápagos region (Figure 7). However, it should be noted here that, because in the currents-only simulation a significant fraction of particles take multiple years to arrive in the Galápagos region, a large part of the downward trend in the left panel in Figure 7 is due to particles having a probability to reach the Galápagos that decreases with time for the last six years of the simulation.

The 'Fate from regional fisheries' scenarios revealed that the probability for particles starting in most of the known fishing locations around the Galápagos to end up on the Galápagos was very small (Figure 8). The total, fishing-hour-weighted fraction of particles that ended up in the Galápagos box was less than 1% for both the currents and currents+wave simulations. Probabilities higher than 5% were only found in fishing locations north and east of the Galápagos in the currents-only simulation, and along the Ecuadorian and Peruvian coastline in the currents+waves simulation, which was in agreement with the results
from the other two scenarios described before.

## 4  Conclusions and Discussion

We have analysed the pathways of virtual particles representing floating microplastics in two sets of simulations; with currents only and with both currents and waves. It is clear that the inclusion of waves had a major effect on the transport of this plastic, and that especially connections to the Northern Hemisphere are reduced due to the effect of waves. The 'Origin from Galápagos' scenario (Figure 3) revealed that it is extremely unlikely for plastic from anywhere but a relatively local region in the Eastern Tropical Pacific, the coastline of South America, and the Southern Ocean to arrive into the Galápagos region.

It is important to note that the virtual particles in these simulations represent highly idealised plastic only. We did not consider beaching, degradation, sinking nor ingestion of plastic. We also did not consider what happens within the Galápagos region.

The simulations agreed well with the trajectories of the GDP drifters (Fig 1). While 40 drifters is not sufficient to do robust statistical comparison (e.g. van Sebille et al., 2009), the patterns of the drifters show similar patterns as the distributions of the virtual particles, especially for the 'Fate from the South American coastline' wind+currents simulation. Since these drifters have mostly lost their drogues by the time they reach the Western Tropical Pacific Ocean (blue lines in Figure 1), it is indeed expected that waves play a role in the dispersion of the satellite-tracked drifters.

The differences between the currents only and currents+wind simulations thus demonstrates the importance of the inclusion of wind effects on the transport of microplastics (Lebreton et al., 2018; Fraser et al., 2018; Onink et al., 2019). These wind-driven Stokes currents, however, are not routinely incorporated into numerical hydrodynamic models, and in fact are not even well-observed. This may change, however, if the European Space Agency's SKIM concept mission to directly measure surface currents from space is launched (Ardhuin et al., 2018). The research presented here highlights again how important it is to observe Stokes drift on a global scale for the simulation of floating debris.

This project forms part of a wider multi-disciplinary programme involving scholars and research teams in marine biology, ecotoxicology, environmental psychology and archaeology. Working collaboratively, and in partnership with local communities, this collaborative effort is expected to develop a better understanding of the causes and consequences of marine plastic pollution in Galápagos than existed previously. Given the understanding of oceanographic currents, the degree of management and policy instruments available, and iconic status of Galápagos, the archipelago is well, even uniquely positioned to provide a demonstration of how a marine reserve can manage and reverse its marine plastic burden. The hope is also that the processes, methodologies, management tools and partnerships established in Galápagos can be extended to other places around the world. Understanding how currents and waves carry plastic from points of deposition ('taps') to places of accumulation ('sinks') is vital. By combining this understanding with the results of other approaches can bring additional insight. For example, an archaeological methodology being trialled in Galápagos uses 'object biographies' or 'life stories' to create narratives around individual items collected from beaches in the archipelago (Schofield, 2018; Schofield et al., in press) to help understand how they got there.

Fieldwork conducted in May and November 2018 involved collecting a representative sample of plastic items from a beach on San Cristobal. These items were then examined in a series of 'Science to Solutions' workshops involving academics and members of the local community, with the aim of building narratives around the coded and visual information each object

contains. The coded information typically includes details of place and date of origin, and the original content (of containers), while visual inspection can betray length of exposure, for example through signs of bleaching and colonisation by marine life.

Preliminary results from the workshops can be compared to the results of the analyses reported here. Most plastic objects found on the beaches were of west-coast South American origin, with many bearing Peruvian and Ecuadorian labels, in agreement with the modelling here. In terms of the objects with Asian labels recorded on the beaches, the results are less clear. It is suspected these objects had not been in the sea for long when they 'landed' in Galápagos, as all are very fresh. This latter observation accords with the results from the finding in this study that items released in Asia would not reach Galápagos. From the object biography workshops, the suggestion instead was that these items were coming from nearby fishing boats originating in SE Asia. This conclusion however is hard to reconcile with the results of the oceanographic modelling, that only a very small percentage of plastics from areas known to be popular fishing grounds would reach the archipelago. Working collaboratively, these very different disciplines and methodologies therefore illustrate both the benefits and some of the challenges of cross-disciplinary and cross-sector partnership to help understand (if not resolve) the challenge of marine plastic pollution.

*Code and data availability.* All scripts that were used to run the simulations are available at https://github.com/OceanParcels/GalapagosBasinPlastic, and the trajectory files are at http://doi.org/10.24416/UU01-5JUDNV. The Parcels code is available at http://oceanparcels.org. The Elipot et al. (2016) Global Drifter Program drifter data is available at ftp://ftp.aoml.noaa.gov/phod/pub/buoydata/hourly_product/v1.02/. The NEMO hydrodynamic data are available from http://opendap4gws.jasmin.ac.uk/thredds/nemo/root/catalog.html. The WaveWatchIII Stokes drift data are available from ftp://ftp.ifremer.fr/ifremer/ww3/HINDCAST/GLOBAL/. The Fishing effort data from Global Fishing Watch (Kroodsma et al., 2018) are available at https://globalfishingwatch.org/datasets-and-code/

*Video supplement.* Animations of the six simulations described here are available as Supplementary material to this manuscript.

*Author contributions.* EvS devised the study, analysed the results of the simulations, and led the writing of the manuscript. PD and EvS ran the Parcels simulations. All authors participated in the writing and editing of the manuscript.

*Competing interests.* No competing interests are present

*Acknowledgements.* This work was supported through funding from the European Research Council (ERC) under the European Union's Horizon 2020 research and innovation programme (grant agreement No 715386) and the European Space Agency (ESA) through the Sea surface KInematics Multiscale monitoring (SKIM) Mission Science (SciSoc) Study (Contract 4000124734/18/NL/CT/gp). BDH is supported by CSIRO Oceans and Atmosphere. The Science to Solutions Workshops were co-hosted by University de San Fransico de Quito Galápagos

Science Centre and The Charles Darwin Research Station. Some of the simulations were carried out on the Dutch National e-Infrastructure with the support of SURF Cooperative (project no. 16371). This study has been conducted using EU Copernicus Marine Service Information. We thank Nicoleta Tsakali for fruitful discussion on preliminary simulations with other models in this context, and Mikael Kaandorp for providing the code for the fisheries simulation.

5  This is part of a multidisciplinary project which involves marine biologists (Ceri Lewis, Adam Porter and JJ, University of Exeter; Juan Pablo Muñoz, University of San Francisco, Quito; Kathy Townsend, University of the Sunshine Coast; Richard Thompson, University of Plymouth, Denise Hardesty, Commonwealth Scientific and Industrial Research Organisation, Australia), a conservation scientist (Brendan Godley, University of Exeter), an ecotoxicologist (Tamara Galloway, University of Exeter), environmental psychologists (Sabine Pahl, University of Plymouth and Kayleigh Wyles, University of Surrey), an archaeologist (JS), and a physical oceanographer (EvS). It is coordinated
10  by the Galapagos Conservation Trust, through AD and JJ (now also at University of Exeter). In addition to many of those people listed above, the workshop described in this paper involved significant participation from the Charles Darwin Research Station and the Galápagos Science Centre in collaboration The Directorate of the Galápagos National Park.

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

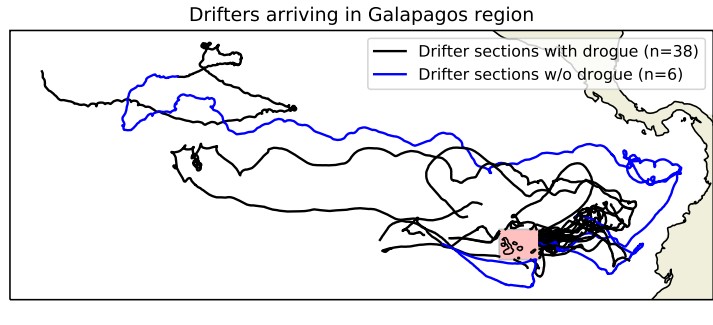

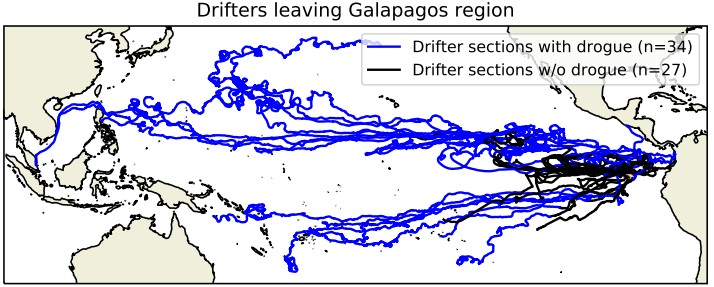

**Figure 1.** Trajectories of surface drifters in the real ocean, from the GDP program (Elipot et al., 2016). Top panel shows drifter trajectories before they arrive in the Galápagos region. Bottom panel shows drifters after they leave the Galápagos region. Black sections of the drifter trajectories indicate when the drifters still have their drogue attached, in the blue sections these drogues are lost.

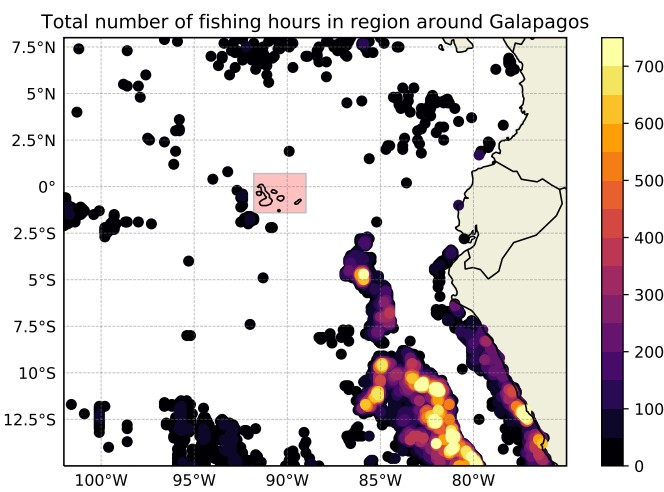

**Figure 2.** Map of locations where, according to the Global Fishing Watch data set from Kroodsma et al. (2018), there was more than 24 hours of Fishing Effort. Circles are color-coded to the total amount of fishing hours in the dataset. Red rectangle denotes the Galápagos region as used throughout this study.

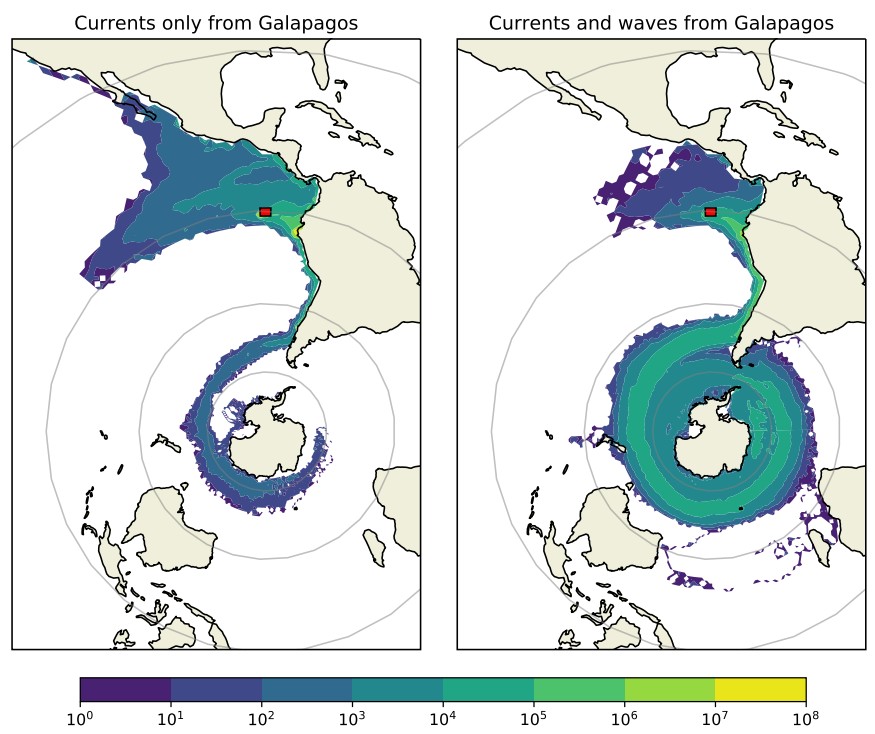

**Figure 3.** Histogram of 'Origin from Galápagos' scenario, showing the density of particle trajectories that end up in the Galápagos region (red rectangle) for particles carried by currents only (top panel) and for particles carried by the currents and waves (bottom panel). The scale is the number of particle crossings per $1° \times 1°$ grid cell, on a logarithmic scale. Gray circles denote the $60°$S, $30°$S, Equator and $30°$N latitude bands. Beaching is not taken into account in this simulation, and maximum length of the trajectories is 10 years. Most trajectories remain in the eastern tropical Pacific Ocean or originate from the Southern Ocean.

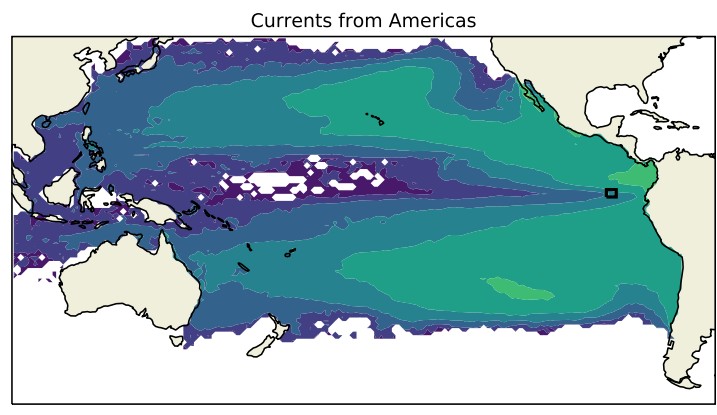

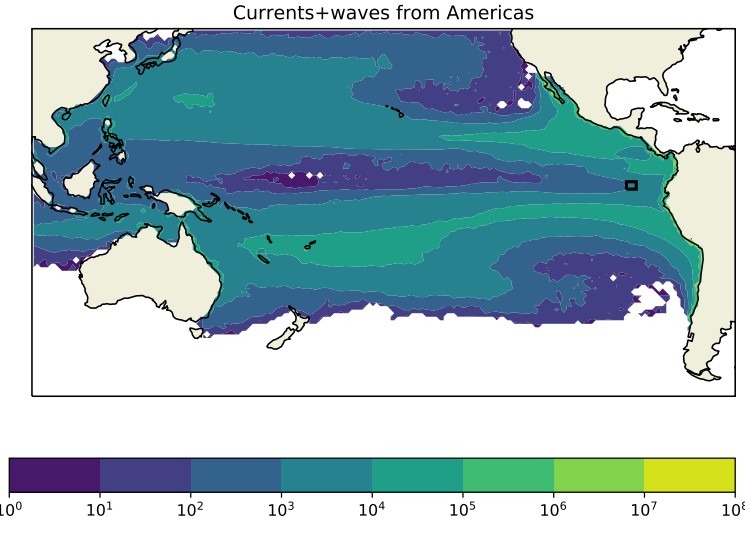

**Figure 4.** Histogram of the 'Fate from the South American coastline' scenario, showing the density of particle trajectories that start on the western coast of the Americas, on a logarithmic color scale, for particles carried by currents only (top panel) and for particles carried by the currents and waves (bottom panel). Maximum length of the trajectories is 5 years. Most particles end up in one of the subtropical gyres, and the Galápagos (black square) is at a relative minimum in both simulations.

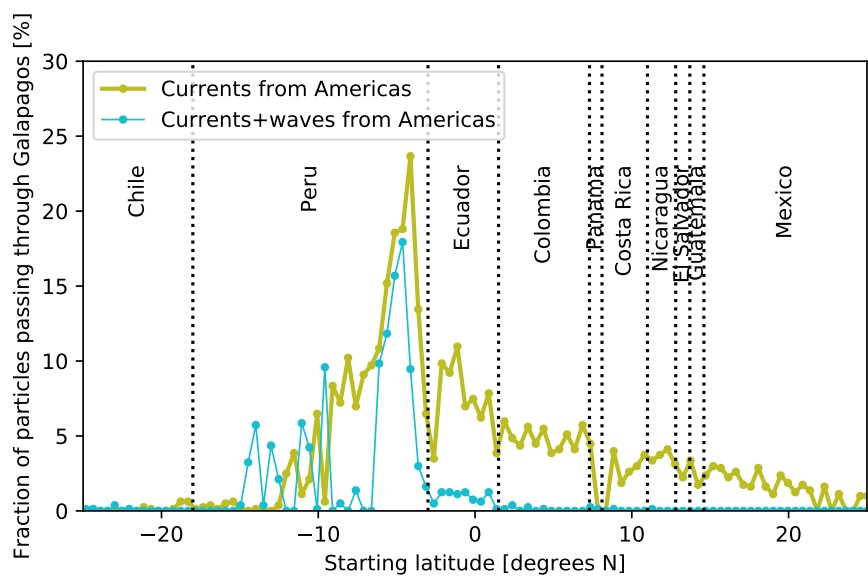

**Figure 5.** The fraction of particles that pass through the Galápagos box as a function of starting latitude for the 'Fate from the South American coastline' scenario, for particles carried by currents only (yellow line) and for particles carried by the currents and waves (blue line). Dashed lines denote the approximate boundaries of different countries along the west-American coast. Most particles that pass through Galápagos start from Northern Peru and Southern Ecuador.

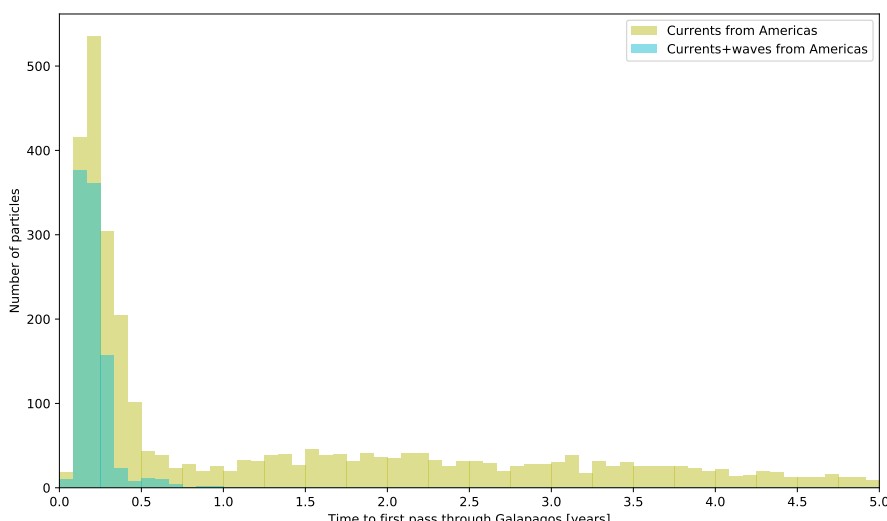

**Figure 6.** Histogram of the time in days required for particles to travel from the west coast of America to the Galápagos region, for particles carried by currents only (yellow bars) and for particles carried by the currents and waves (blue bars). Most particles arrive within 3-4 months, although there is a significant tail all the way to 5 years for the simulation with currents only.

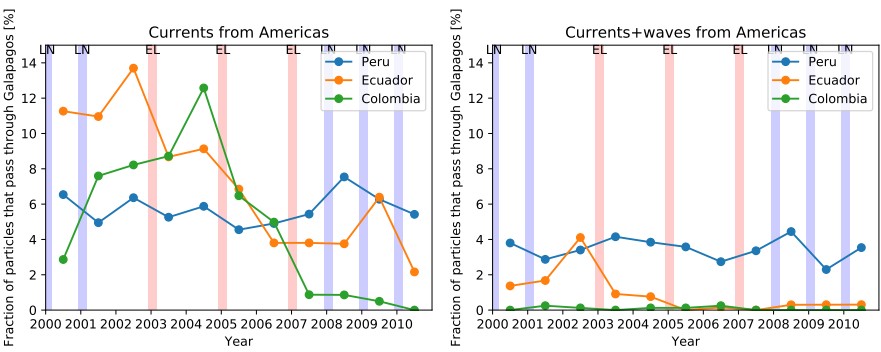

**Figure 7.** Time series of the fraction of particles starting in Peru, Ecuador and Colombia that pass through the Galápagos region, for particles carried by currents only (left panel) and for particles carried by the currents and waves (right panel). Blue bars indicate La Niña periods, red bars indicate El Niño periods. While there is no apparent relation between ENSO state for Peru and Ecuador, it is clear that the fraction of particles carried by currents only that end up in the Galápagos region from Colombia is much higher during El Niño than during La Niña periods.

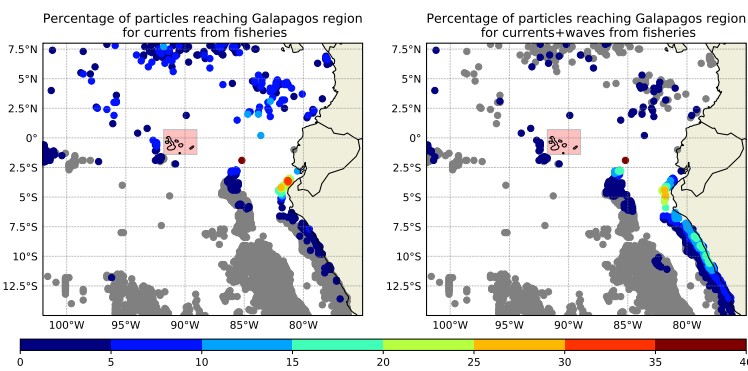

**Figure 8.** Maps from the 'Fate from regional fisheries' scenario, showing the percentage of particles that reach the Galápagos region (red box) from each of the 3,885 locations where at least 24 hours of fishing was reported in the Global Fishing Watch dataset (Kroodsma et al., 2018). Left panel shows percentages for the currents-only simulation, right the percentages for the currents+wave simulation. Floating particles from most of these locations have a zero probability of ending up near the Galàpagos within 5 years (grey circles), but there are extensive regions of non-zero probabilities (coloured circles) near the Peruvian and Ecuadorian coast.