# Peer review of "Basin-scale sources and pathways of microplastic that ends up in the Galápagos Archipelago"

_Ocean Science, 2019_

## Referee Comment (RC1) · Anonymous Referee #1 · 2 Jul 2019

This is a interesting manuscript that I think is worth publication. I have two rather minor points.

I think the manuscript would benefit from a more full discussion regarding comparison/analysis of the forward and backward trajectory analysis.

Also, I would appreciate that the forward model is run for a longer time and spans a larger area for initiation. Plastic has entered the ocean for a long time, and the two year trajectory analysis is a bit short. At least the restriction in the analysis for using a two year simulation should be considered.
* * *

---

## Referee Comment (RC2) · Anonymous Referee #2 · 10 Jul 2019

Major comments:

The paper address the transport of plastics at basin-wide scales from land-based origins to the Galapagos Islands, considering high resolution simulations that also include Stokes' Drift wave contributions to particle advection. The paper illustrates global environmental analysis needed for conservation efforts at the Galapagos. The paper is well written, but could benefit from some additional scientific detail and discussion.

Discussion of Stokes' Drift contribution, and its affect on Lagrangian particles, could be more developed as part of the literature review. Given the prominent role of Stokes' Drift on the results, it would benefit from greater up front introduction and discussion. Overall the paper addresses a novel use of Lagrangian particles needed for environmental protection, although use of forward and backward trajectories is less clear unless the

backward in time, forward in time process is non-hysteretic within some error tolerance. Discussion of such the error tolerance would help provide additional confidence in the results.

Note, this is a nice example of how science can clear up a mystery related to plastics arriving at the Galapagos- could consider playing this up a little more to make the paper even more interesting than it already is given the need for ecological preservation in key sites like the Galapagos.

Below are some specific recommendations on ways this nice paper can be improved.

Pg 3, line 10 Please comment on why (or why not) 5-day temporal resolution is sufficient

Pg 3, line 115 - Please detail how currents+waves are used to advect particles. This isn't clear

Pg 3, line 19-20- How do you know you have enough particles to make the inference? Would you get the same answer with twice as many or few particles?

Png 4, line 10-12. Why do waves contain particles close to their release location? This is stated but it isn't clear why this may be the case. Line 17- which ones are better?

Pg 4, line 23- how consistent are the forward / backward runs? If you do a backward run to get a particle initial position and then do a forward run, does the particle return to its initial position? Is a discussion of backward / forward time particle hysteresis warranted? If not, why not?

Pg 4, line 28- do particles decay also? If so, this may be a good place to highlight or discuss this potential issue too.

Minor comments:

Pg 2, Line 8: "natural values in 1978" to "natural value in 1978"
[Figure]

Pg 3, line 30 - "currents+waves" would be better described as "combined waves and current flow"

Pg 4, line 3 - nice reference and connection to broader work

Figures:

Figures are of good quality although text labels in Figure 4 could be rotated vertically for clarity.

Figure 5 implies that mixing changes as a function of Stokes' Drift contribution to particle advection. These plots could be used to derive advective and dispersive timescales for the transport, which would be a good way to more quantitatively make result comparisons.

---

## Author Comment (AC1) · 5 Aug 2019

*Reviewer 1:*

*This is a interesting manuscript that I think is worth publication. I have two rather minor points. I think the manuscript would benefit from a more full discussion regarding comparison/ analysis of the forward and backward trajectory analysis.*

Following also the comments from reviewer 2, we have been more careful in our discussion of the forward and backward scenarios. The key difference is that one scenario analyses the fate of plastic, while the other scenario analyses the origin of plastic. Because release scenarios are also very different, they are not one-on-one comparable. We have now clarified this in the manuscript by explicitly calling the scenarios 'Origin

from Galápagos' and 'Fate from the South American coastline'. We have also added some further discussion of the hysteresis to section 3 (page 5, lines 19-21 of the track-changed manuscript).

*Also, I would appreciate that the forward model is run for a longer time and spans a larger area for initiation. Plastic has entered the ocean for a long time, and the two year trajectory analysis is a bit short. At least the restriction in the analysis for using a two year simulation should be considered.*

We have now redone all the forward analyses tracking particles for 5 years. See Figure 1 for the histogram maps, which shows that more particles reach the Western Tropical Pacific and leak into the Indian Ocean, but differences in the Eastern Pacific are relatively minor.

In fact, there are no extra particles that reach the Galápagos region if the simulations are run for 5 years in the currents+waves simulations, something that was already expected from the original manuscript Figure 5, which indicated that all particles that reach the Galápagos do so within one year. However, there are some differences in the currents only simulations, with more particles from the north of the Equator (Figure 2).

Given these (small) differences, we have now updated all forward analysis in the manuscript to the 5-year simulations and changed the figures and text accordingly.

Please also note the supplement to this comment:
https://www.ocean-sci-discuss.net/os-2019-37/os-2019-37-AC1-supplement.pdf
* * *
Currents 2 years      Currents+waves 2 years

Currents 5 years      Currents+waves 5 years

$10^0$   $10^1$   $10^2$   $10^3$   $10^4$   $10^5$   $10^6$   $10^7$   10     $10^1$   $10^2$   $10^3$   $10^4$   $10^5$   $10^6$   $10^7$   $10^8$

**Fig. 1.** Histogram of the 'Fate from the South American coastline' simulation particle trajectories, for 2-year long simulations (top row) and 5-year long simulations (bottom row).

![Figure 2: plot of fraction of particles passing through Galapagos versus starting latitude, comparing Currents 2 years and Currents 5 years]

**Fig. 2.** Comparison of the fraction of particles that pass through the Galápagos box for the currents-only simulation of lengths 2 years (original manuscript) and 5 years (revised version). Note that there was

**Supplement:**

[revised manuscript text omitted]

---

## Author Comment (AC2) · 5 Aug 2019

*Reviewer 2:*

*Major comments: The paper address the transport of plastics at basin-wide scales from land-based origins to the Galapagos Islands, considering high resolution simulations that also include Stokes' Drift wave contributions to particle advection. The paper illustrates global environmental analysis needed for conservation efforts at the Galapagos. The paper is well written, but could benefit from some additional scientific detail and discussion.*

We thank the reviewer for their careful and useful comments. Below, we detail how we have addressed each of them

*Discussion of Stokes' Drift contribution, and its affect on Lagrangian particles, could be more developed as part of the literature review. Given the prominent role of Stokes' Drift on the results, it would benefit from greater up front introduction and discussion.*

We have now extended the paragraph on Stokes drift in the Introduction (page 2, lines 31 and further in the revised manuscript): "There is still a debate in the community to what extent wave-induced currents – so-called Stokes drift (Stokes, 1847) – has an impact on the transport of plastic (Lebreton et al 2018; Onink et al 2019), so we analyzed the particle pathways both with and without this effect of waves. Stokes drift is the net drift velocity in the direction of wave propagation experienced by a particle floating at the free surface of a water wave (see van den Bremer and Breivik (2018) for a recent review). Its magnitude is generally much smaller than that of the surface currents (e.g. Figure 1 of Onink et al 2019), but because Stokes drift has large spatial coherence patterns its long-term effect on particle transport can be significant (Fraser et al 2018)."

*Overall the paper addresses a novel use of Lagrangian particles needed for environmental protection, although use of forward and backward trajectories is less clear unless the backward in time, forward in time process is non-hysteretic within some error tolerance. Discussion of such the error tolerance would help provide additional confidence in the results.*

Following also the comment from Reviewer 1, we have been more careful in our discussion of the forward and backward simulations. The key difference is that one analyses the fate of plastic, while the other the origin of plastic. Because release strategies are also very different, they are not one-on-one comparable. We have now clarified this in the manuscript by explicitly calling the simulations 'Origin from Galápagos' and 'Fate from the South American coastline'. We have also added some further discussion of the hysteresis to section 3 (page 5, lines 19-21 of the track-changed manuscript).

*Note, this is a nice example of how science can clear up a mystery related to plastics*

*arriving at the Galapagos- could consider playing this up a little more to make the paper even more interesting than it already is given the need for ecological preservation in key sites like the Galapagos.*

We thank the reviewer for this very nice comment. Following this idea, we discussed how to make the manuscript even more policy-relevant and realised that plastic from fisheries is another potential source that requires information for adequate management. In the revised version, we therefore added a new scenario (Figures 2 and 8) where we analysed the fraction of particles released from known fishing locations (from the Global Fishing Watch dataset) that reach the Galápagos region.

*Below are some specific recommendations on ways this nice paper can be improved.*

*Pg 3, line 10 Please comment on why (or why not) 5-day temporal resolution is sufficient*

We have now added (page 3, lines 16-18 in the track-changed manuscript) that "as Qin et al showed that time-averaging errors are small for temporal resolutions shorter than 9 days in a 1/10° spatial resolution, this 5-day temporal resolution is sufficient."

*Pg 3, line 115 - Please detail how currents+waves are used to advect particles. This isn't clear*

We clarified (page 3, lines 25-27 in the track-changed manuscript) that "Parcels v2.0 has inbuilt support for advection of particles on two different Fields using Summed-Field objects, so that the velocities at each location are interpolated and then summed at each RK4 substep (see also Delandmeter and van Sebille, 2019), and the currents+wave simulations were done with that feature."

*Pg 3, line 19-20- How do you know you have enough particles to make the inference? Would you get the same answer with twice as many or few particles?*

We have redone some of the analysis using only 50

We have added some lines to the manuscript (page 4, lines 3-4 in the track-changed manuscript): "Redoing all the analyses below with only half of the particles does not affect the results and conclusions, giving us confidence that we released sufficient particles."

*Png 4, line 10-12. Why do waves contain particles close to their release location? This is stated but it isn't clear why this may be the case. Line 17- which ones are better?*

We have now clarified that this is because the wave-induced Stokes drift is generally eastward near the South American coast, leading to a coastward transport by the waves (page 5, line 6 of the track-changed manuscript). And for the difference between forward and backward scenarios, it is not a question of which one is 'better'. Rather, their interpretation can be used to answer different questions. The forward scenarios can help answer the question 'where does floating plastic from this location end up?', while the backward scenarios can help answer the question ('where does plastic found here come from?'). Both questions are important, which is why we discuss them both in our manuscript.

*Pg 4, line 23- how consistent are the forward / backward runs? If you do a backward run to get a particle initial position and then do a forward run, does the particle return to its initial position? Is a discussion of backward / forward time particle hysteresis warranted? If not, why not?*

As we now discuss in the manuscript (page 3, lines 6-11 in the track-changed manuscript), the difference between the Fate and Origin scenarios is not so much hysteresis in the forward and backward simulations, but rather the choice of particle seeding locations and the interpretation of the simulations. We have now clarified that in the manuscript

*Pg 4, line 28- do particles decay also? If so, this may be a good place to highlight or discuss this potential issue too.*

We have clarified that the simulations don't take fragmentation of particles into account, because the fragmentation rates are very poorly constrained by observations (page 5, lines 26-28 in the track-changed manuscript: "Longer residence times in the ocean will also likely lead to more fragmentation, but this is also not taken into account because the time scales involved are very poorly constrained from observations (Cològar et al., 2014)."

*Minor comments:*

*Pg 2, Line 8: "natural values in 1978" to "natural value in 1978"*

This has been fixed in the revised manuscript

*Pg 3, line 30 - "currents+waves" would be better described as "combined waves and current flow"*

We have now added in section 2 (page 3, lines 23-25 in the track-changed manuscript) that we advect particles in either "only the NEMO surface flow fields (hereafter referred to as the 'currents' simulations), or the combined NEMO surface flow and WaveWatchIII Stokes drift fields (hereafter referred to as the 'currents+waves' simulations)"

*Pg 4, line 3 - nice reference and connection to broader work*

We thank the reviewer for this positive comment

*Figures: Figures are of good quality although text labels in Figure 4 could be rotated vertically for clarity.*

We have now rotated the text labels in Figure 5 of the revised manuscript. Note that some are still difficult to read, but that is because the coastline for e.g. El Salvador is so small. Given that the text labels are only for guidance and are directly related to the latitude anyways, we do not think this is a major problem.

*Figure 5 implies that mixing changes as a function of Stokes' Drift contribution to particle advection. These plots could be used to derive advective and dispersive timescales*

*for the transport, which would be a good way to more quantitatively make result comparisons.*

We agree with the reviewer that the 'effective diffusion' of Stokes drift is an interesting topic by itself, but a thorough analysis of this effect would require a completely different simulation setup from the one used here (with much more uniform seeding, so that particulars of regional flow can be averaged out). We'd therefore prefer to do that in a separate study, not particularly focused on the Galápagos region.

Please also note the supplement to this comment:
https://www.ocean-sci-discuss.net/os-2019-37/os-2019-37-AC2-supplement.pdf

Currents from Americas

Currents from Americas

Currents+waves from Americas

Currents+waves from Americas

$10^0$ $10^1$ $10^2$ $10^3$ $10^4$ $10^5$ $10^6$ $10^7$ $10^8$ $10^1$ $10^2$ $10^3$ $10^4$ $10^5$ $10^6$ $10^7$ $10^8$

**Fig. 1.** Figure 4 from the revised manuscript (left) and the same analysis using only a random selection of 50% of the particles (right).

[Figure]

[Figure]

**Fig. 2.** Figure 5 from the revised manuscript (left) and the same analysis using only a random selection of 50% of the particles (right).

**Supplement:**

[revised manuscript text omitted]